# How to Evaluate the Green and High-Quality Development Path? An *FsQCA* Approach on the China Pilot Free Trade Zone

**DOI:** 10.3390/ijerph19010547

**Published:** 2022-01-04

**Authors:** Guanglan Zhou, Zhening Zhang, Yulian Fei

**Affiliations:** 1Modern Business Research Center, Zhejiang Gongshang University, Hangzhou 310018, China; guanglanzhou@zjgsu.edu.cn; 2Academe of Zhejiang Culture Industry Innovation & Development, Zhejiang Gongshang University, Hangzhou 310018, China; 3School of Management Engineering and E-Commerce, Zhejiang Gongshang University, Hangzhou 310018, China; fyl@zjgsu.edu.cn

**Keywords:** green development, pilot free trade zone, development path, fuzzy-set qualitative comparative analysis, configuration analysis

## Abstract

In today’s world, green development has become an important trend, and many countries regard the development of green industry as an important measure to promote economic restructuring. Green development is consistent with sustainable development in ideology. China’s economy is in the stage of high-quality development. As an important foundation for China’s external economic development, the free trade zone can play a good role in promoting its green and high-quality development. Based on the data of 18 free trade zones in China in 2020, this paper explores the green and high-quality development path of the China Pilot Free Trade Zone. Firstly, the green development index is constructed according to the existing research and experience, and then the fuzzy set qualitative comparative analysis method is used to evaluate the green and high-quality development path. The results show that the development of pilot free trade zones is not the result of a single condition but the result of a combination of green policy effectiveness, foreign investment participation, green production growth rate, and other conditions. Combined with the green and high-quality development path, this paper further provides enlightenment for the development of the China Pilot Free Trade Zone.

## 1. Introduction

Global climate change is an indisputable fact and has become one of the biggest challenges to human development in the 21st century. Anthropogenic factors of climate change are mainly caused by economic activity since the industrial Revolution. With the rapid development of China’s economy, energy use and carbon dioxide emissions are also increasing dramatically. Since 1750, the global cumulative emissions of 1.1 trillion tons of carbon dioxide, developed countries accounted for 80% of the emissions, the United States accounted for 26.9% in the first place, and China ranked second, accounting for 8.2% [1]. In addition, according to the latest figures, China has overtaken the United States as the world’s largest annual emitter of carbon dioxide. As the effects of climate change intensify, China is already facing increasing pressure from the world to reduce emissions. As a matter of fact, China’s energy conservation and emission reduction is not only a response to international pressure but also an inherent requirement of China’s economic development mode transformation. During China’s development, it has put forward the path of sustainable development, scientific development, and green development. Sustainable development means that we should consider both the needs of current development and those of future development, instead of meeting the interests of present generations at the expense of future generations [2]. Scientific development is people-oriented, comprehensive, coordinated, and sustainable development [3]. Green development is a model innovation based on traditional development. It is a new development model that takes environmental protection as an important pillar to achieve sustainable development under the constraints of ecological environment capacity and resource carrying capacity [4]. The three paths are progressive and closely linked. Scientific development is a necessary condition for sustainable development. Scientific development contains the spirit of sustainable development, and green development is an important part of scientific development, which is more specific. From sustainable development to scientific development and then to green development, addressing global climate change and energy conservation and emission reduction are both challenges and opportunities for China. China needs to seize the opportunity and blaze a path of green development.

China put forward the development strategy of pilot free trade zones in 2013 and set up its first pilot free trade zone in Shanghai. So far, China has set up 21 free trade zones, covering two-thirds of the country. According to the data of The Ministry of Commerce, in 2020, the total import and export trade of the 18 free trade zones established in the first five batches reached 4.7 trillion-yuan, accounting for 14.7% of the total import and export of the country. The development of free trade zones plays a significant role in regional economic development. High-quality development is a new expression first put forward by China in 2017 [5], indicating that the Chinese economy is shifting from a stage of high-speed growth to a stage of high-quality development. In the context of China’s regional strategy of “Four sectors + Three economic belts”, the construction of free trade zones should also pay more attention to high-quality development, to promote high-quality development of regional economy. At the same time, low-carbon action has become a global action that affects the common destiny of mankind and is an inherent requirement for harmonious coexistence between man and nature. High-quality development embodies a new vision of development, and green and low-carbon development is a distinctive feature of high-quality development. Achieving carbon peak is a major strategic decision of the government and an inherent requirement for green and high-quality development [6].

Traditional industry, especially the heavy industry, has demonstrated its damage on the earth. In addition, thus, the global supply chain has moved to the developing countries. However, the developing countries, such as China, are reluctant to be polluted by the divergence of the industry chains from other developed nations. It is necessary to explore a novel path for these nations. Digital trade presents the opportunity to China. Through the digitalization, the global supply chain could be formed as the low-carbon emission, environment friendly development. Therefore, this paper objectively and reasonably evaluates the development path of the China Pilot Free Trade Zone and finds an effective path suitable for the green and high-quality development of the China Pilot Free Trade Zone, which is conducive to the vigorous development of China’s regional economy. Focusing on the basic requirement that experience can be replicated and extended, with green as the background and characteristics, we will lead China’s free trade zones to deepen reform, opening-up and institutional innovation, and contribute wisdom and solutions to the green and high-quality development of global free trade zones.

## 2. Literature Review

There are few research results on the green and high-quality development of free trade zone. Most scholars tend to evaluate and analyze the free trade zone in a certain region, and there are few studies on the path planning of the development of the free trade zone from the national level, and the analytic hierarchy process and fuzzy comprehensive evaluation method are mainly used to establish the index system. No scholars have applied fuzzy set qualitative comparative analysis method to evaluate and study the green and high-quality development path of free trade zone. Fuzzy set qualitative comparative analysis method is very suitable for path evaluation research. In combination with the current international scholars’ research content on free trade zone, the application of FsQCA method, and relevant literature on the development status of the China Pilot Free Trade Zone, this paper decided to use FsQCA method to explore the green and high-quality development path of the China Pilot Free Trade Zone.

### 2.1. Free Trade Zone

Ricardo’s theory of comparative advantage points out that comparative advantage is the basis of trade between countries, and each country or region should concentrate on the production of products with comparative advantages and improve labor productivity through trade with other countries or regions, which is also the source of the theoretical basis of free trade zones [7]. The establishment of free trade zone is an important means for a country to gain advantages in the process of adapting to globalization and developing global trade [8]. Since the concept of free trade zone was put forward, scholars have focused on the policy effect of free trade zone, innovative development, and regional economic growth. From the perspective of the innovation and development of the free trade zone, the establishment of the free trade zone has a positive effect on the improvement of regional innovation capacity [9]. Some scholars believe that the construction of free trade zones can introduce foreign capital and increase import and export trade, thus promoting the development of regional economy [10,11,12]. At the same time, a good institutional environment of the free trade zone can promote knowledge output and provide a foundation for industrial agglomeration and innovation level improvement [13,14]. In addition, some scholars have found a strong connection between the establishment of the free trade zone and production performance [15,16,17], the change of enterprise performance in the free trade zone will affect the development of the free trade zone, and the policy effect of the free trade zone will attract enterprises, thus affecting the organization and performance of enterprises [18].

Since the reform and opening, China has taken a series of measures to promote foreign investment and trade, such as the establishment of special economic zones [19]. China’s first free trade zone was established in the 1980s and was manufacturing-export-oriented [20]. In the following period, China’s manufacturing advantage was gradually replaced by low-cost competitors in Southeast Asia [21]. However, after China’s entry into the world trade organization, the Shanghai Free Trade Zone was first founded in September 2013, and this move helped China to eliminate fortress ownership restrictions on foreign investment and services; at the same time, force shall be protected by law of service companies achieve modernization [21]. The establishment of the free trade zone not only meets China’s own development needs but also addresses the concerns of foreign investors caused by the growing international dissatisfaction with China’s trade development. The establishment of the Shanghai Free Trade Zone is China’s response to the U.S.-led Trans-Pacific Partnership [22]. However, free trade zones also face many problems, China’s free trade zones have made some achievements in physical trade and e-commerce, and many enterprises have entered the free trade zones through the opportunity of deregulation in the free trade zones. However, the overall innovation level of China’s free trade zones is low, and they are highly dependent on policies, and have no obvious advantages or functions [17]. From the perspective of international trade, China is “big but not strong”. China’s free trade zones lack innovation, product quality, added value, and integration in trade in goods and services [23].

To put it simply, the benefit of free trade zone lies in the unrestricted circulation of goods from various countries and regions, so that consumers can choose a wide range of goods in the market, and free trade is zero tariff, which can benefit consumers. In addition, it also can alleviate the adverse effects of monopoly market to a certain extent and promote the diversification of world economy. However, free trade zones also have disadvantages for host countries. Under the influence of free trade, the disadvantages of a country’s industrial structure will gradually come out, which will present a great hindrance to the development of the whole country’s trade. In order to better develop and better participate in the international market trade competition, it is necessary to actively adapt to the changes of various market environment factors brought by free trade, so the original industrial pattern with increasing abuses must be effectively adjusted. However, the long-formed industrial pattern cannot be easily changed overnight. In order to accelerate the active adjustment and improvement of the industrial pattern, it is bound to be supported by the input of the government from the material capital, which is also a considerable cost consumption. In addition, if a country is weak in macroeconomic regulation of its economy, the implementation of free trade will, to a certain extent, cause the loss of its own economic rights. Because, under the background of free trade, all kinds of economic capital, than before on the circulation, are more convenient and quicker, so the economic flow rate also can appear as rapid ascension; to a certain extent, this can disrupt a national macroeconomic regulation and control of the original rhythm, making the adjustment of the economic structure and capital operate in a certain degree of confusion, and the acquisition of the original economic rights will cause different degrees of damage.

### 2.2. Green Development

In this article, green development refers to the concept of green development proposed by China. The government put forward five development concepts of “innovation, coordination, green, open, and sharing” and made green development an important concept of China’s development. Green development requires that environmental protection should be considered, while pursuing economic development, and the quality and efficiency of development should be attached importance to sustainable development.

Ecological civilization may overcome the most fundamental double bind of global capitalism in the 21st century—the long-standing tension between economic development and human sustainability [24]. Earlier studies explored the relationship between carbon emissions, financial development, trade status, energy consumption, and economic growth in the context of Turkey [25]. Some scholars studied the importance of green manufacturing and the methods of mitigating resource and environmental problems and found that green manufacturing was originally intended to reduce costs and protect the environment. Therefore, they proposed that the construction of green manufacturing system could reduce resource waste, provide favorable environment for product development, and promote green industrial development [26]. It is found that green manufacturing can improve environmental and social performance by studying the sustainable development performance of enterprises from three dimensions: economy, environment, and society [27]. Using a fuzzy multicriteria decision-making model for manufacturing green transformation has found that enterprises have a competitive advantage, and the government enacted incentive measures and technology will promote industry green transformation of the distribution of resources [28]. Through a questionnaire survey, someone explored the mechanism model of guiding tourists’ sustainable consumption intention based on environment and consumption behavior and provides corresponding strategies for consumers to further promote the development of green economy [29]. In the field of e-commerce, some scholars have proposed an effective POI (point-of-interest) recommendation method based on preference, social relations, and temporal and spatial factors, to improve the accuracy of recommendations, alleviate the problems of cold start and sparsity of recommendations, and provide theoretical and practical support for the development of green economy [30,31]. The concept of green development is currently considered to encompass economic systems, natural systems, and social systems, namely a model of social development in which environmental economic growth is not achieved at the expense of the environment and people’s health [32].

## 3. Methods

Fuzzy-set stereotyped comparative analysis (FsQCA) is usually suitable for studies covering 5–50 cases [33]. FsQCA uses Boolean algebra to analyze example of cases as variables and result arrangements [34]. The combination of appropriate sample size and different conditional variables makes FsQCA a suitable analytical tool for analyzing the different paths of the leading outcome variables [35]. Using FsQCA can get “different shades of gray” for variables [33]. FsQCA 3.0 software can assist in the analysis of necessary conditions and sufficient conditions, first necessary conditions, then sufficient conditions [36]. After Ragin [35], a condition is considered necessary to indicate that any outcome occurs when it occurs, although existence does not ensure that it occurs. When a condition is defined as sufficient, it indicates that it is observed in all cases and occurs when the result occurs.

FsQCA has been used in many fields. In the field of information systems, it is found that there are five effective ways to achieve high level network governance in PSN governance practice [37]. In the field of online business and marketing, by combining the factors of trust, privacy, emotion, and experience, this paper observes the changes of customers’ purchase intention, and then develops a new emotion-centered theory and the design of providing personalized service and proposes a step-by-step method for how to apply FsQCA in e-commerce research [38]. In the field of consumer psychology, fuzzy set qualitative comparative analysis is used to identify the complex preconditions of some individual contradictory belief behaviors in the field of social and environment-oriented behaviors and orientations [39]. The field of strategy and organizational research includes explaining how qualitative comparative analysis can be used to enrich the configuration theory of strategy and organization and contribute to configuration research [40]. In the field of education, the causal patterns of factors affecting students’ continued study of computer science were explored, and eight configurations of cognitive and non-cognitive gains, barriers, learning motivation, and learning performance were found to explain the high willingness to continue study of computer science.

In this paper, fuzzy set qualitative comparative analysis method (FsQCA) is used to analyze the combination of variables, and the generation path and influencing factors of the green and high-quality development of China’s free trade zone are analyzed from the perspective of configuration. The specific reasons for the adoption of FsQCA method in this paper are as follows: FsQCA method is suitable for medium and small sample measurement analysis. At present, the free trade zones established in China are still in the stage of innovative policy experiment. The number of free trade zones established is small, which belongs to the category of small sample measurement and meets the conditions of fuzzy set qualitative comparative analysis. Compared with the traditional regression analysis method, FsQCA can find the configuration relations and paths of various factors. According to the characteristics of complex and diverse influencing factors in the development of free trade zone, it is necessary to carry out the configuration analysis of different factors. FsQCA has more advantages than other types of QCA analysis techniques (such as csQCA and mvQCA).

### 3.1. Data, Sample, and Variates

As of October 2021, China has set up 21 free trade zones, covering 2/3 of the provinces of the country. As the new free trade zones set up after 2020 are in the stage of initial development and policy implementation, data are missing. However, the policy implementation of the free trade zones set up in 2019 and before is relatively perfect, and the policy effect is relatively obvious. Therefore, based on the data integrity and the development degree of the free trade zones, this paper decides to select 18 pilot free trade zones established by the end of 2019 as research objects, and Table 1 shows the specific spatial distribution. The data of 2020 were selected as the data source of the article through the statistical Yearbook, statistical bulletin, and official websites of each free trade zone.

In terms of specific analysis techniques, the result variable of this paper is the comprehensive score of the development quality of the free trade zone, which is a continuous variable ranging from 0 to 1. FsQCA method can more fully discover the influence of antecedent condition changes, improve data accuracy, and better avoid information loss in the process of data processing. Combined with the actual situation of the development of China’s free trade zone, this paper sets antecedent conditions for the evaluation of the development quality of free trade zone from five dimensions of green industrial upgrading ability, green policy effectiveness, foreign participation, green production growth rate, and trade development level, which conforms to the research premise of FsQCA method.

#### 3.1.1. Outcome Variables

Development quality of free trade zone: In order to achieve sustained and stable development and bring economic benefits to the region, the free trade zone must pursue green and high-quality development. At present, China is vigorously exploring the path of green development, and most industries have completed green transformation and upgrading. The development quality index is an important index reflecting the comprehensive benefits of the economic development of the free trade zone and a hard index based on public service products. Therefore, this paper selects the comprehensive development quality indices of 18 free trade zones in 2020 as the outcome variables, which is according to the development quality index constructed by Xu, and the data of 2020 were imported to obtain the final result [41].

#### 3.1.2. Conditional Variables

The green and high-quality development of the free trade zone is influenced by the external environment and the internal environment. Through reasonable planning and perfect strategies, the free trade zone can gradually move towards the road of green and high-quality development.

At present, China’s foreign trade dependence is higher, to a large extent influenced by the development level of regional economy in China, and industrial upgrading comprehensive performance is the quality of the regional industry development, through the establishment and development of free trade zone to attract industrial agglomeration, to rely on free trade area within the scale development to promote regional economic development. Green industry refers to the industry that strives to save resources and reduce pollution (energy saving and emission reduction) by means of a green production mechanism based on environmental protection consideration in the production process. With the development of the Internet, it now covers digital trade developed through the Internet. To comply with the requirements of green development, the green industrial upgrading ability is selected as one of the conditional variables; specifically, the growth rate of investment in green projects within the industry is used to reflect the upgrading level of green industry.

The free trade zone is the central and local policies in China, which implement the achievements inspection, the relevant policies to a certain degree reflecting the size of the regional economic and trade, and driving and regional policy enforcement is the main source of quality free trade area development. Policies related to low carbon and environmental protection have provided great support for the green development of the free trade zone, and taking the green policy effectiveness as one of the conditional variables can better analyze the impact of relevant policies on the green and high-quality development of the free trade zone. The Chinese government hopes to develop green free trade zones, vigorously promote trade in high-quality and high-value-added green products, strengthen international cooperation on green standards, better align green trade rules with import and export policies, deepen cooperation on green Belt and Road initiatives, and expand cooperation on technology, equipment, and services in energy conservation, environmental protection, and clean energy. The added value of green capacity in the secondary and tertiary industries can well reflect the impact of green policy on the industry.

Due to the development of the free trade zone, there is a close relationship with the area of the open economy, foreign capital utilization of quality has important influence on the development of free trade zone, and it has a scholar in the study of the problem of location selection of free trade zone in China to introduce foreign capital participation and explore its effect on free trade zone construction. Some scholars have conducted an empirical study on the impact of foreign direct investment on China’s green total factor productivity by using Chinese provincial panel data from 2001 to 2012 and FGLS method, and the results show that foreign direct investment plays a significant role in promoting the growth of China’s green total factor productivity. So, the research on development factors also need to consider the foreign participation, and with the actual use of foreign capital amount to do specific expression.

When using QCA method to study the location choice of Chinese enterprises’ OFDI, it is found that production growth rate has an important external effect on the location choice of FDI. Green, low-carbon, and high-quality development has achieved healthy GDP growth in the region. At present, China’s free trade zones are greatly improving the degree of green production by developing production methods with high technological content, low resource consumption and less environmental pollution, and promoting the formation of a green production system. The growth rate of green production in the free trade zone can reflect the quality of green development in the free trade zone. Therefore, the green production growth rate is selected as the factor affecting the quality of green development of the free trade zone in this paper, which is expressed by the production growth rate of the part related to green growth in the regional GDP growth rate.

In the free trade zone development quality evaluation, the influence of the degree of market opening should also be considered. The government fully recognized the importance of open markets and fair trade; therefore, taking trade transformation as an important measure is conducive to providing data support for the path study of the development quality of free trade zones. This paper decided to choose the trade development level as a condition of variable, which is expressed by the increased in total import and export trade and the increased turnover of cargo.

To sum up, FsQCA method is adopted in this paper to evaluate and study the green and high-quality development path of China’s free trade zones by selecting the five factors of green industrial upgrading ability, green policy effectiveness, foreign participation, green production growth rate, and trade development level of 18 free trade zones. The main description of the causal conditions and outcome variables before the development quality of China’s free trade zone is shown in Table 2.

Data in 2020 were selected from the Statistical Yearbook, Statistical Bulletin, and official websites of each free trade zone. The specific data of indicator variables of China’s 18 free trade zones in 2020 are shown in Table 3.

### 3.2. Measure Calibration

In FsQCA, each condition (green industrial upgrading ability, green policy effectiveness, foreign participation, green production growth rate, and trade development level) and outcome (development quality of free trade zone) are, respectively, regarded as a set. Each case has a membership score in the set. Calibration is the process of giving a membership score to the case. Calibration is usually carried out according to existing theoretical knowledge or experience. Since there are few studies on the development quality of free trade zones, there is no prior theoretical or practical knowledge as the basis for calibrating antecedent and outcome variables. Therefore, referring to previous studies, this paper adopts the quartile method to directly calibrate variables. According to the data type of each condition and result, this paper uses the direct calibration method to convert the data into membership fraction of fuzzy set. Three anchor points were determined: “complete membership = 75 points”, “turning point = 50 points”, and “complete non-membership = 25 points”. The calibration parameters are shown in Table 4.

### 3.3. Analytical Steps

#### 3.3.1. Necessity Analysis of Univariate

This paper first tests whether a single condition (including its non-set) constitutes a necessary condition for the green and high-quality development of the free trade zone. The consistency is used to measure the necessary condition. When the consistency level is greater than 0.9, the condition is the necessary condition for the result. Table 5 shows the test results of the necessary conditions for the green and high-quality development of the free trade zone. Table 5 shows that all variables (including its not set) on the consistency of the free trade zone development quality evaluation are less than 0.9. Therefore, there are no necessary conditions for the green and high-quality development of free trade zones in Table 5’s analysis results of the necessary conditions. It indicates that any single factor in the five variables cannot lead to green and high-quality development of the free trade zone. Further combinatorial analysis of the antecedent conditions is required.

#### 3.3.2. Sufficiency Analysis of Conditional Configuration

Configuration analysis is different from necessary condition analysis in that it aims to analyze the adequacy of results resulting from different configurations consisting of multiple conditions. Adequacy analysis is measured by consistency index. In this paper, the consistency threshold is set as 0.8, and the case frequency threshold is set as 1. FsQCA software is used to output three kinds of solutions with different complexity: complex solution, reduced solution, and intermediate solution. In this paper, the intermediate solution is used to distinguish the core conditions, and the reduced solution is used to distinguish the core variables and auxiliary variables of grouping states. According to Ragin (2008) and Fiss (2011) [42,43], a solid circle indicates the existence of the condition, a cross circle indicates the absence of the condition, a blank space indicates a fuzzy state (the condition can exist or be absent), a large circle indicates the core condition, and a small circle indicates the auxiliary condition. The resulting configuration results in a green and high-quality development path are shown in Table 6.

As can be seen from Table 6, there are four paths that can lead to green and high-quality development outcomes, with consistency indicators ranging from 0.845238 to 0.980392, which meet Woodside’s criteria [44]. The overall consistency is 0.820645, indicating that the explanatory degree of the four configurations to the evaluation of the green and high-quality development level of China’s free trade zone is about 82.06%. The overall coverage is 0.898305, indicating that 89.83% of cases can be covered by the empirical analysis results in this paper. The consistency level of both the single solution (configuration) and the overall solution is higher than 0.8, indicating that all configurations have a good explanatory degree for the result variables. Since FsQCA method can explain the problem of causal asymmetry, this paper obtains two paths leading to the non-high-quality development of free trade zone through analysis, and the two configurations also meet the standards of the consistency test.

### 3.4. Robustness Tests

In this study, the method of adjusting the consistency level was adopted for the robustness test to ensure the stability of the green and high-quality development configuration of the free trade zone. Without changing the case frequency, the consistency level was increased by 0.05 for the robustness test. Table 7 is the result. It can be found that there is no change between the tested configuration and the original configuration, which proves that the research conclusion of this paper has good robustness.

## 4. Discussion

### 4.1. Green and High-Quality Development Configuration

Through data analysis, this paper obtained four configurations leading to the green and high-quality development results of free trade zone: Configuration A(fsPE∗~fsFP∗~fsMZ∗~fsTL); Configuration B1(fsPE∗fsFP∗fsMZ∗fsTL); Configuration B2(~fsOA∗fsFP∗fsMZ∗~fsTL); and Configuration C(fsOA∗~fsPE∗fsMZ), and the following gives a detailed description of the four green and high-quality development paths (configuration) of the free trade zone. The sample cases covered by the above configuration and the number are shown in Table 8:

#### 4.1.1. Configuration A(fsPE∗~fsFP∗~fsMZ∗~fsTL)

The core condition in this configuration is green policy effectiveness, indicating that this condition plays a core role in the development of the free trade zone, while foreign capital participation and trade development level are auxiliary conditions due to the lack of obvious advantages. The possibility of existence of this architecture is 0.862876, and the unique coverage is 0.118710, which can explain 11.87% of green and high-quality development, covering Hainan, Tianjin, and Heilongjiang, with the largest number of cases. This path can lead to green and high-quality development results in the free trade zone.

According to the geographical location and regional development of Hainan Free Trade Zone, the country gives its unique positioning. As the only area in China to bring the whole province into the scope of the free trade zone, it has been favored by many national policies, as the fourth batch of a free trade zone, and Hainan made good reference to the development experience of the previously established free trade zone, and promoted the green and high-quality development of the free trade zone through constant adjustment and innovation in the pilot process, relying on the application and coordination of national policies. The Tianjin Free Trade Zone, established in the second batch, is an important node of China’s “Belt and Road” construction plan and the China-Mongolia-Russia Economic Corridor, and the free trade zone, since its establishment by speeding up the construction of the “Internet + governmental affairs service” platform, to trade enterprises raises a lot of convenience, made through which decentralization reform government functions shift gradually to the service, further improve the business environment, and promote the high quality development of the Tianjin Free Trade Zone. As early as 2018, General Secretary Xi Jinping emphasized at the symposium on In-depth Promotion of Northeast Revitalization that northeast China should strengthen the construction intensity of the free trade zone and vigorously promote all-round revitalization of northeast China. Since its establishment, the Heilongjiang Free Trade Zone has copied and promoted 10 policy achievements of other free trade zone. Good policy and system basis is the prerequisite for the introduction of various industries. At the same time, as an important window connecting Russia’s Far East and opening to the north, under the joint efforts of the Chinese and Russian governments, the transportation facilities have been constantly improved, providing great convenience for bilateral trade. The above three free trade zones achieved green policy effectiveness in different ways and also achieved green and high-quality development results.

#### 4.1.2. Configuration B1(fsPE∗fsFP∗fsMZ∗fsTL)

The core conditions in this configuration are foreign participation and green production growth rate, with the trade development level and green policy effectiveness as auxiliary functions. This path shows that high participation of foreign investment and high green production growth rate are sufficient conditions for green and high-quality development of the free trade zone. The probability of the existence of configuration B1 is 0.980392, and the unique coverage is 0.087742, which can explain 8.77% of green and high-quality development, covering the two cases of Guangdong and Zhejiang. This path can lead to green and high-quality development results of the free trade zone.

Since reform and opening up, the pearl river delta region relies on system innovation and geographical advantages, by attracting foreign investment and fostering local industry to build a “world manufacturing industry base”, which offers free trade zone development in Guangdong after its industrial foundation; at the same time, in the free trade agreement signed between Guangdong, Hong Kong, and the framework of CEPA, the cooperation between Guangdong promoted the liberalization of trade in services in Guangdong. Further expanding the market scale has laid a good foundation for the operation of the Guangdong Free Trade Zone. The Hengqin zone takes advantage of its geographical location and environmental advantages to introduce high-quality resources from Hong Kong, Macao, and the world to develop high-tech industries. In Zhejiang province, the industrial concentration area as a high quality of foreign capital is concentrated; to create outstanding main body function, source of foreign investment is a relatively concentrated industry international garden, vigorously attracting high-end industry to invest in Zhejiang. Ningbo-Zhoushan port is the world’s largest comprehensive port, and the T junction is located in the economy, and the goods quantity is large, helping to strengthen the international market in China in commodity storage and transportation of trade. These two free trade zones also achieved green and high-quality development results by increasing foreign participation and green production growth rate in different ways.

#### 4.1.3. Configuration B2(~fsOA∗fsFP∗fsMZ∗~fsTL)

The core condition of this configuration is the same as that of configuration B1, which are both foreign participation and green production growth rate. This configuration indicates that the combination of foreign participation and green production growth rate can make up for the deficiency of green industrial upgrading ability and trade development level. The probability of existence of this configuration is 0.845238, and the unique coverage is 0.135484, which can explain 13.55% of green and high-quality development, covering Shanghai and Shandong.

The Shanghai Free Trade Zone is China’s first set up free trade zone, and its development level has also gained a lot of attention. Since the end of the 20th century, Shanghai has attracted foreign direct investment by relying on the influence of various special economic zones, resulting in a high level of foreign investment participation after the establishment of the Shanghai Free Trade Zone. At the same time, the Shanghai free trade zone actively guides foreign investors to set up multinational headquarters in the free trade zone, provides support for enterprises meeting relevant conditions to develop offshore trade business, and encourages integrated development of enterprises. Under the influence of the Shanghai Free Trade Zone, the economy of the surrounding cities has also developed rapidly, and it is more convenient to realize regional economic development at the same time. By combining its own economic advantages with the business opportunities brought by the Shanghai Free Trade Zone, the Yangtze River Delta region has better reduced trade costs, expanded market scale, and developed its internal and external trade. The Shandong Free Trade Zone, as the fifth batch of free trade zone construction, has clearly pointed out in its overall plan to provide favorable treatment for knowledge and knowledge-based talents. At the same time, innovative incubation centers, Chinese-foreign joint universities, and industrial technology research institutes have been set up to attract many outstanding talents domestically and abroad to the Shandong Free Trade Zone. By attracting talents, introducing advanced technology, and foreign investment, the Shandong Free Trade Zone will form a strong endogenous economic driving force with strong radiation capacity, thus promoting the green and high-quality development of the free trade zone itself. These two free trade zones have achieved green and high-quality development by increasing foreign capital participation and expanding green production growth rate in different ways.

#### 4.1.4. Configuration C(fsOA∗~fsPE∗fsMZ)

The core condition in this configuration is green industrial upgrading ability, indicating that this condition plays a core role in the development of the free trade zone, and the green production growth rate is an auxiliary condition to compensate for the lack of green policy effectiveness. The possibility of existence of this architecture is 0.845238, and the unique coverage is 0.135484, which can explain 13.55% of green and high-quality development. Only Hebei is covered, and the number of cases covered is the least. This path of free trade zone can also lead to green and high-quality development results.

The establishment of the Hebei Free Trade Zone in 2019 has brought fresh impetus to the coordinated development of The Beijing-Tianjin-Hebei region. The Hebei Free Trade Zone takes advantage of its four zones with different functions and complementary advantages to attract relevant high-tech industries to gather in each zone. The Xiongan zone focuses on the development of information technology, biotechnology, and ecological agriculture, the Zhengding zone is mainly engaged in high-end equipment manufacturing and international logistics and transportation, the Caofeidian zone is a place of energy reserve and international commodity trade, and the Daxing airport zone according to the geographical position is mainly engaged in aviation logistics and aviation technology. The establishment of the Hebei Free Trade Zone brings advantages for industrial transformation and upgrading and optimization of resource allocation. The development of the free trade zone to promote the development of surrounding areas, shorten the gap between Beijing and Tianjin, and promote the coordinated development of the three places will aid in achieving the green and high-quality development of the free trade zone.

### 4.2. Non-High-Quality Development Configuration

The analysis results show that there are two types of non-high-quality development configuration; one is the green production growth rate restraint type, and the other is the green industrial upgrading ability, trade development level restraint type, and the combination of these two types of conditional variables cannot achieve green and high-quality development of the free trade zone. By comparing with the six types of paths of the development quality level of the free trade zone, it can be found that the factors affecting the evaluation of the development quality of the free trade zone are typically asymmetrical, i.e., the reasons promoting the green and high-quality development of the free trade zone are not completely opposite to the reasons inhibiting the green and high-quality development of the free trade zone. Therefore, we cannot sum up the reasons for the green and high-quality development and the non-high-quality development of the free trade zone as an either/or single relationship.

## 5. Conclusions

### 5.1. The Green and High-Quality Development of the Free Trade Zone Has Multiple Concurrent Characteristics and Follows the Principle of All Road’s Leading to the Same Destination

It is a new attempt to explain the green and high-quality development of free trade zones from the perspective of configuration and overall. The green and high-quality development of free trade zones can be caused by green policy effectiveness dominant type, green industrial upgrading ability dominant type, and foreign capital participation-market size dominant type. The combination configurations of different antecedent conditions have complete equivalence without conflict. When the core conditions affecting green and high-quality development do not exist, the emergence of other influential factors can still effectively lead to green and high-quality development results. Therefore, the green and high-quality development of the free trade zone is not the only combination of specific factors and conditions, but there are diversified paths and scenarios, following the principle of all road’s leading to the same destination.

### 5.2. There Is Causal Asymmetry between Green and High-Quality Development and Non-High-Quality Development of Free Trade Zone

The combination of antecedent conditions of green production growth rate restraint and green industrial upgrading ability-trade development level restraint is difficult to make the free trade zone achieve green and high-quality development. The green and high-quality development of the free trade zone can be the result of green policy effectiveness, the linkage and matching of foreign capital participation and trade development level, or the effect of green industrial upgrading ability. All these paths will lead to the green and high-quality development of the free trade zone. Therefore, there is causal asymmetry between green and high-quality development evaluation mechanism and non-high-quality development evaluation mechanism.

### 5.3. Implication for China’s Pilot Free Trade Zone

According to the research results of this paper, no single factor can promote the green and high-quality development of the free trade zone. Therefore, to make China’s free trade zone present a green and high-quality development level, it is necessary to select an appropriate path according to the characteristics of regional development by combining the antecedent conditions, such as green industrial upgrading ability, green policy effectiveness, and green production growth rate.

First, actively explore the development path of green and high-quality free trade zones with Chinese characteristics. Through pilot-driven development, institutional innovation, and policy guidance, we should reduce the carbon emission intensity of free trade zones and take the lead in reaching the peak of carbon emissions, give full play to the leading and demonstration role of the free trade zone with green and high-quality development, promote the transformation of trade development mode, drive the low-carbon industry, technology, products, and services through the low carbonization of trade, and improve the international competitiveness of China’s green industry. Our country can provide reference and replicable successful experience for the low-carbon development of other regions through pioneering experiments.

Second, fully recognizing the long-term nature and complexity of developing green and high-quality free trade zones, while considering the needs of economic growth and stable trade development, is also important. Other aims are focus on improving the green development capacity of China’s free trade zone, focus on building the carbon emission management system of the free trade zone, integrate the concept of green development into the development planning of the free trade zone, and effectively promote the green development of the free trade zone itself. With the existing policy system, at the same time, through the free trade zone of low carbon and green development policy for effective docking, important are building the low carbon emissions standards, low carbon green industrial policy support and financial investment to guide industry development policy system, through building green low carbon industry chain, and improve the international competitiveness of China’s low carbon products and services exports.

Third, an aim should be to continue to promote the green transformation and upgrading of the industrial structure of the free trade zone, strictly control products with low added value and high energy consumption from the source, vigorously introduce green development projects, and increase the proportion of high added value products and services in the export product mix. Industrial access standards will be raised, with emphasis on encouraging and supporting the development of strategic emerging industries, such as solar photovoltaic industry, electronic information industry, modern service industry, and producer service industry in the free trade zone, to promote the overall optimization of the industrial structure in the free trade zone.

Fourth, promoting the green development of export trade of products and services will strengthen the overall green and low-carbon supply chain of the zone and improve the competitiveness of green and low-carbon products and services of the free trade zone in the international market. Through fiscal subsidies, tax breaks, and other preferential policies, enterprises are encouraged to develop and produce low-carbon products, and consumers are guided to buy green products to promote the transformation of the whole society to green and low-carbon. This will promote the establishment of a green management system covering the whole industrial chain of free trade zones, including product research and development, product design, product packaging, clean production, green circulation, and green marketing.

With the construction and development of pilot free trade zones, China has not only realized the full coverage of coastal provinces but also extended inland and realized the layout of pilot free trade zones in border areas, forming economic growth poles and improving the level of regional economic development. With the upgrading of industries in different regions, the inter-regional industrial integration is further deepened. The resources and labor advantages in the central and western regions and the capital, technology, and talent advantages in the eastern regions complement each other in the industrial integration development, thus promoting the coordinated development of regional economy. To further strengthen the green development of the pilot free trade zone and promote high-quality development of the pilot free trade zone, strengthening green trade is an important way and starting point to promote high-quality development of the pilot free trade zone and fully reflects the leading role of the pilot free trade zone in the development of green trade. The green and high-quality development of the FREE trade zone will provide new impetus for the high-quality development of China’s economy.

## Figures and Tables

**Table 1 ijerph-19-00547-t001:** Establishment and spatial distribution of 18 free trade zones in China.

Region	Time of Establishment	Regional Orientation	Regional Strategy
Shanghai	2013.09	Eastern region	Yangtze River Economic Belt, Maritime Silk Road
Guangdong	2015.04	Eastern region	Guangdong-Hong Kong-Macao Greater Bay Area, Maritime Silk Road
Tianjin	2015.04	Eastern region	Coordinated Development of the Beijing-Tianjin-Hebei Region, Bohai Rim Economic Belt
Fujian	2015.04	Eastern region	Maritime Silk Road
Liaoning	2017.03	Northeastern region	Revitalize the old industrial base in northeast China, Silk Road Economic Belt
Zhejiang	2017.03	Eastern region	Yangtze River Economic Belt, Maritime Silk Road
Henan	2017.03	Central region	The Rise of Central China
Hubei	2017.03	Central region	Yangtze River Economic Belt
Chongqing	2017.03	Western region	Yangtze River Economic Belt, Silk Road Economic Belt
Sichuan	2017.03	Western region	Yangtze River Economic Belt
Shaanxi	2017.03	Western region	Silk Road Economic Belt
Hainan	2018.04	Eastern region	Maritime Silk Road
Shandong	2019.08	Eastern region	Bohai Rim Economic Belt
Jiangsu	2019.08	Eastern region	Yangtze River Economic Belt
Guangxi	2019.08	Eastern region	Silk Road Economic Belt
Hebei	2019.08	Eastern region	Coordinated development of The Beijing-Tianjin-Hebei region
Yunnan	2019.08	Western region	Silk Road Economic Belt, Yangtze River Economic Belt
Heilongjiang	2019.08	Northeastern region	Revitalize the old industrial base in northeast China

**Table 2 ijerph-19-00547-t002:** Description of outcome variables and antecedent condition indicators.

Indicator		Acronym	Description
Outcome variables	Development quality	DQ	Comprehensive development quality index
Conditional variables	Green industrial upgrading ability	UA	Growth rate of fixed asset investment for green industries
	Green policy effectiveness	PE	The value added by the green development policy to the secondary and tertiary industries
	Foreign participation	FP	Amount of foreign capital actually utilized
	Green production growth rate	GR	Regional GDP growth rate due to green development
	Trade level	TL	Total import and export trade increased, Cargo turnover

**Table 3 ijerph-19-00547-t003:** Index variable data of free trade zone.

Region	Growth Rate of Fixed Asset Investment for Green Industries (%)	The Value Added by the Green Development Policy to the Secondary and Tertiary Industries (¥100,000,000)	Amount of Foreign Capital Actually Utilized ($1000)	Regional GDP Growth Rate Due to Green Development (%)	Total Import and Export Trade Increased (%)	Cargo Turnover (Tons·km)	Comprehensive Development Quality Index
Hainan	−9.2	4228.57	152,021	5.8	6.8	1648.03	71.22
Zhejiang	10	60,254.36	1,998,100	6.8	8.2	12,391.92	59.26
Liaoning	0.3	22,731.68	332,292	5.5	−4	8921.43	52.90
Henan	8	49,623.8	1,872,700	7	3.7	8658.54	47.53
Tianjin	13.1	13,919.05	473,200	4.8	−9.1	2662.45	76.67
Hubei	10.7	42,019.22	1,290,700	7.5	13.2	6132.4	47.54
Chongqing	5.6	22,054.35	236,500	6.3	11	3614.15	52.39
Shaanxi	2.5	23,802.24	589,400	6	0.1	3482.15	52.06
Shanghai	5.1	38,051.44	1,904,800	6	0.1	30,324.9	81.75
Sichuan	8.9	41,808.58	1,247,854	7.5	13.8	2710.83	51.41
Guangdong	11.1	103,319.81	2,294,800	6.2	−0.2	27,373.67	60.76
Fujian	5.9	39,798.77	1,587,000	7.6	7.8	8292.13	46.99
Shandong	−8.4	65,951.09	1,468,933	5.5	5.8	10,166.42	57.16
Jiangsu	5.1	95,335.24	2,612,425	6.1	−0.9	9947.68	51.85
Guangxi	9.6	17,849.4	110,946	6	14.4	3989.18	50.95
Hebei	6.5	31,586.08	1,028,000	6.8	12.6	13,563.38	57.31
Yunnan	8.5	20,186.13	72,300	8.1	12.8	1552.05	52.15
Heilongjiang	6.3	10,430.24	54,000	4.2	6.7	1615.08	53.20

**Table 4 ijerph-19-00547-t004:** Result variables are calibrated with antecedent conditions.

Results and Conditional Variables	Complete Membership	Turning Point	Complete Non-Membership
DQ	63.635	52.63	50.3325
UA	9.7	6.4	4.45
PE	52,281.44	34,818.76	19,601.9475
FP	1,880,725	1,137,927	215,380.25
GR	7.125	6.15	5.725
TL	10,722.795	7212.265	2698.735

**Table 5 ijerph-19-00547-t005:** Analysis results of necessary condition.

Antecedent Conditions	Green and High-Quality Development		Non-High-Quality Development	
	Consistency	Coverage	Consistency	Coverage
UA	0.52841	0.98934	0.53200	0.16049
~UA	0.57326	0.82394	0.97600	0.24314
PE	0.45677	0.88950	0.63900	0.18376
~PE	0.57178	0.92357	0.57700	0.13556
FP	0.52198	0.93842	0.65200	0.18789
~FP	0.53812	0.84890	0.74900	0.21358
GR	0.34288	1.00000	0.18400	0.08336
~GR	0.67462	0.84728	1.00000	0.19926
TL	0.66309	0.83940	0.63200	0.12131
~TL	0.33942	0.85738	0.51200	0.23940

~ means logical “not”, that is, not subject to the antecedent condition.

**Table 6 ijerph-19-00547-t006:** Green and high-quality development architecture of free trade zone.

Antecedent Conditions	Green and High-Quality Development	Non-High-Quality Development
A	B1	B2	C	d	e
UA			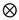	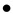		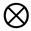
PE	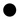	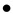		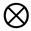		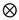
FP	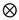	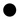	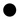		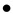	
GR	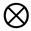	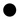	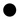	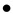	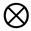	
TL	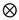	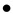	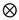		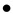	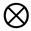
Original coverage	0.38492	0.33489	0.17384	0.25930	0.38940	0.29847
Unique coverage	0.11948	0.088352	0.13904	0.16489	0.21149	0.25398
Consistency	0.86732	0.98221	0.84579	0.86773	0.87462	0.93477
Consistency of the Population solution	0.83236	0.82945
Coverage of the population solution	0.89881	0.63374

Note: ‘
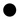
’ indicates that the condition exists and is the core condition. ‘
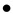
’ that the condition exists and is an auxiliary condition. ‘
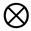
’ indicates that the core condition does not exist; ‘
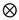
’ indicates that the auxiliary condition does not exist. The space indicates an ambiguous state.

**Table 7 ijerph-19-00547-t007:** Results of the robustness test of green and high-quality development.

Antecedent Conditions	Green and High-Quality Development
A	B1	B2	C
UA			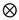	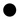
PE	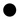	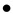		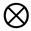
FP	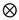	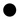	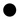	
GR	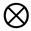	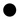	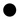	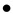
TL	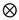	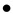	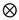	
Original coverage	0.38492	0.33489	0.17384	0.25930
Unique coverage	0.11948	0.088352	0.13904	0.16489
Consistency	0.86732	0.98221	0.84579	0.86773
Consistency of the Population solution	0.83236
Coverage of the population solution	0.89881

Note: ‘
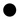
’ indicates that the condition exists and is the core condition. ‘
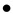
’ that the condition exists and is an auxiliary condition. ‘
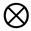
’ indicates that the core condition does not exist; ‘
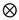
’ indicates that the auxiliary condition does not exist. The space indicates an ambiguous state.

**Table 8 ijerph-19-00547-t008:** Sample cases and number of configurations covered above.

Configuration	Sample Cases Covered	Number
A	Heilongjiang (0.96, 0.57); Hainan (0.94, 1); Tianjin (0.91, 1)	3
B1	Zhejiang (0.87, 0.94); Guangdong (0.54, 0.96)	2
B2	Shandong (0.79, 0.55); Shanghai (0.66, 1)	2
C	Hebei (0.53, 0.87)	1

## Data Availability

The data used to support the findings of this study can be obtained from the China Statistical Yearbook, the Statistical Bulletin, and the official website of the China Free Trade Zone.

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
