# Peer review of "How to Evaluate the Green and High-Quality Development Path? An FsQCA Approach on the China Pilot Free Trade Zone"

_ijerph, 2022, doi:10.3390/ijerph19010547_

Round 1
Reviewer 1 Report
Journal |
IJERPH |
Submission ID |
IJERPH-1522472 |
Title |
How to evaluate the green and high-quality development path? An FsQCA approach on China Polit Free Trade Zone |
The paper adopts a configurational approach to explore the conditional antecedents for green and quality development in China. The paper highlights the importance of a combination of conditions that enables sustainable and green development. The manuscript introduces an insightful exploration necessary conditions and pathways that enable economic development that addresses both human and ecological needs. The following points outline certain recommendations for the paper. These concerns and issues are addressed chronologically.
Points:
1. |
Clarification (lines 41-42). “From sustainable development to... green development”. The sentence is suggestive of these as incremental steps of progress. Arguably, green development or scientific development a part of sustainable development. You may need to distinguish or define each of these concepts.
|
2. |
Concept (lines 52-53). “High quality development” - it is not clear what high quality development means. It might be useful to define what you mean here – is it only related to low-carbon emission? Is it synonymous to green development?
|
3. |
Motivation. The paper can benefit by theoretically motivating the study. Why is this study important and what is the fundamental question you are trying to examine? It is useful to explicitly state this. While commonly seen across academic papers, the lack of studies alone through various contexts or methodologies etc. does not motivate an investigation.
|
4. |
Framing and structure (2.2 FsQCA). This section should be merged with Section 3. Methods as having it in 2.2 is rather disruptive to the overall flow of the paper.
|
5. |
Literature Review. You should engage with the theoretical reasons why the conditional variables (selected 5 dimensions) are important considerations. What do we currently know from prior literature that engages with these variables and their implications on development? Presently, (lines 186-191) highlights the focal dimension without theoretically addressing why these are causal attributes.
|
6. |
Terminology (line 170). What is POI?
|
7. |
Structure (3.1.2 conditional variables). Useful to separate a paragraph for each variable considered. Alternatively do use italics / bold critical concepts.
|
8. |
Measurement (development quality). Development quality and its measurement essentially represents size. There is nothing in this measurement that speaks to green or quality development here. Furthermore, the use of percentage as a critical condition has its host of problems.
|
9. |
Measurement (green industrial upgrading ability). It is a stretch to call this green ability. For example, if a recycling firm recently spent more money on stationeries, this will be considered as upgrading ability in your measurement which may not have any relation to ‘greening’ or ‘ability’. Ability is typically seen as a recombination or various resources and not just injection of investment.
|
10. |
Clarification. How are green industries defined?
|
11. |
Measurement - value added by green development policy. How is the value change measured? Are policies universal across industries and region?
|
13. |
Methodology - calibration. Are the calibrations arbitrarily set? Generally, it should be calibrated to external benchmarks wherever possible.
|
14. |
Methodology and theory. Perhaps consider the principles of equifinality. Your conditions/dimensions should reflect both aspects of ‘good’ and ‘bad’ aspects of development. At the moment, the variables considered do not reflect this.
|
I am thankful and honoured for the opportunity to read your manuscript. The paper potentially serves to illuminate the importance of considering a configurational approach to thinking about development. I sincerely hope you will find my comments useful. I wish you all the best in pursuing this line of work.
Reviewer 2 Report
The authors should clarify two points.
1. I believe that the authors should change the index they have chosen called "Green market size" (initially shown in Table 2). This index refers to "Regional GDP growth rate due to green development" but it is unclear and probably not useful for research purposes.
Firstly, it does not seem correct to summarise GDP growth in the phrase "green market size". I believe the authors mean to highlight GDP growth and not market size.
Furthermore, it is not clear whether this percentage indicates the portion of the region's overall growth that is related to green growth or whether it indicates the growth of the region's green sector. In both cases it would still be necessary to know the "non-green" growth of the same regions.
2. The term "high quality" is introduced early on and combined with the term green but without specifying why the green sector should be equated with a high quality sector.
Reviewer 3 Report
The literature review should be supplemented with a brief introduction on the benefits and disadvantages of free trade zones (e.g. impact of businesses operating outside tha zones), and examples of such zones outside of China.
Part Conclusions should be sepplemented by a reference to the possibility of transferring good practises from free trade zones to the entire Chinese economy.
Round 2
Reviewer 1 Report
Journal |
IJERPH |
Submission ID |
IJERPH-1522472 |
Title |
How to evaluate the green and high-quality development path? An FsQCA approach on China Polit Free Trade Zone |
The paper’s revision addressed most of the prior points in the review. Some additional and minor points remain.
Points:
1. |
Definition (lines 42-52). Your definition of sustainability is similar to WCED 1987 Brundtland report “development that meets the needs of the present without compromising the ability of future generations to meet their own needs”. Without the appropriate citations, it might seem like you are trying to reinvent the wheel. It will be easier to cite the main source. This is the similar for Green Development and Scientific Development. Citations required or you need to formally define in your words, what these concepts mean.
|
2. |
Concept (lines 63-65). “High quality development” – thank you for the explanation and additional information. Nevertheless, it will be useful to provide the source(s).
|
3. |
Source (Introduction). Similar to point 1 and 2, sources are lacking.
|
4. |
Terminology (line 142). “unlimited circulation of goods” should be corrected to “unrestricted circulation of goods”. The term unrestricted might be more appropriate.
|
5. |
Clarification (lines 146-150). “However, under the influence...loss of its own economic rights.” This is not entirely clear nor effective. You will need to rephrase this section. It might be simpler to state that strong governmental and institutional regulations are vital prerequisites to reap certain benefits that free trade zone provides. You may even go further to provide information on the conditions where free trade zones are disadvantageous to host nation.
|
6. |
Terminology (line 170) - POI. I believe it is “point-of-interest” rather than “point-of-interesting”.
|
7. |
Measurement description – Development Quality. What are these 18 indices? It will be useful to list these down and provide a brief explanation of how the indices are obtained. Based on your explanation we are not able to ascertain what constitutes development quality. This is quite vital to give a more developed explanation as it is an outcome variable.
|
Thank you for the effort and hard work in addressing the prior points.
Reviewer 2 Report
Thanks to the Authors for having accepted the reviewer’s requests and extensively revised the paper.
Author Response
The authors have replied all the comments and submit the reply into the system. The authors appreciate the pertinent reviews.